# Characterization of Three Types of Elongases from Different Fungi and Site-Directed Mutagenesis

**DOI:** 10.3390/jof10020129

**Published:** 2024-02-03

**Authors:** Yuxin Wang, Lulu Chang, Hao Zhang, Yong Q. Chen, Wei Chen, Haiqin Chen

**Affiliations:** 1State Key Laboratory of Food Science and Resources, Jiangnan University, Wuxi 214122, China; 6210112089@stu.jiangnan.edu.cn (Y.W.);; 2School of Food Science and Technology, Jiangnan University, Wuxi 214122, China; 3National Engineering Research Center for Functional Food, Jiangnan University, Wuxi 214122, China

**Keywords:** fatty acid elongase, characterization, *Mucor circinelloides*, *Phytophthora ramorum*, *Phytophthora sojae*, mutagenesis

## Abstract

Fatty acid elongases play crucial roles in synthesizing long-chain polyunsaturated fatty acids. Identifying more efficient elongases is essential for enhancing oleaginous microorganisms to produce high yields of target products. We characterized three elongases that were identified with distinct specificities: McELO from *Mucor circinelloides*, PrELO from *Phytophthora ramorum*, and PsELO from *Phytophthora sojae*. Heterologous expression in *Saccharomyces cerevisiae* showed that McELO preferentially elongates C16 to C18 fatty acids, PrELO targets Δ6 polyunsaturated fatty acids, and PsELO uses long chain saturated fatty acids as substrates. McELO and PrELO exhibited more homology, potentially enabling fatty acid composition remodeling and enhanced LC-PUFAs production in oleaginous microorganisms. Site-directed mutagenesis of conserved amino acids across elongase types identified residues essential for activity, supported by molecular docking. Alanine substitution of conserved polar residues led to enzyme inactivation, underscoring their importance in the condensation reaction. Our findings offer promising elongase candidates for polyunsaturated fatty acid production, contributing to the bioindustry’s sustainable development.

## 1. Introduction

Long-chain polyunsaturated fatty acids (LC-PUFAs), such as eicosapentaenoic acid (EPA, 20:5 ω-3) and docosahexaenoic acid (DHA, 22:6 ω-3) are essential dietary components [1]. The limited natural sources of LC-PUFAs (such as fish oil and algae oil) cannot satisfy the increasing market demand. Consequently, oil-producing microorganisms, recognized for their safe consumption and high oil content, are utilized for LC-PUFAs production [2]. In previous studies, it has been found that increasing the expression level of elongase has an important effect on the yield of LC-PUFAs [3,4,5,6]. However, LC-PUFAs biosynthesis still encounters many challenges, such as the lack of converting enzymes with higher activity and how to break through the limitation of lipid homeostasis [7]. Thus, identifying and characterizing fatty acid elongases with improved catalytic activity show great promise for advancing research in the food sciences.

In general, LC-PUFAs are synthesized through a series of oxygen-dependent desaturation and elongation reactions. There are multiple membrane-bound elongation systems, which are responsible for the chain extension of fatty acids. The elongase system is composed of four enzymes: a condensing enzyme (β-ketoacyl CoA synthase, KCS), β-ketoacyl CoA reductase (KCR), β-hydroxyacyl CoA dehydrase, and *trans-*2-enoyl CoA reductase [8]. Ketoacyl CoA synthase named fatty acid elongases catalyze the rate-limiting condensation reaction within this complex elongation systems and determines the substrate specificity of the elongation reaction [9].

Elongases with high activity can be isolated from microorganisms which are rich in polyunsaturated fatty acids or endowed with specific fatty acid ratios. *Mucor circinelloides*, the first microorganism commercially exploited for oil production due to its high γ-linolenic acid content, possesses four elongases with yet-to-be-determined functions [10,11]. Genetic modification of many oil-producing microorganisms includes overexpression of elongase to increase C16 fatty acid elongation [3,4]. As a lipid-producing fungus rich in C18 fatty acids, indicating the C16/C18 elongase from *M. circinelloides* may have high activity. Oomycete fungi like *Pythium* sp. and *Phytophthora* sp. can synthesize 20-carbon PUFA, from which a variety of desaturase enzymes have been isolated and characterized, but few elongases have been identified [12]. Elongases and desaturases from *Phytophthora* species have facilitated genetic engineering efforts [3]. Thus, we hope to discover novel or more active elongases from these species. In this study, we identified a C16/C18 elongase in *Mucor circinelloides* (McELO), a Δ6-elongase from *Phytophthora ramorum* (PrELO), and an SFA-elongase from *Phytophthora sojae* (PsELO) through heterologous expression in *Saccharomyces cerevisiae*.

The substrate specificity-determining sites within the elongase catalytic domain remain experimentally unvalidated, hindering further enzyme modification and application. Recent studies have utilized various methods to investigate functional regions of elongases, including sequence comparison to identify conserved amino acid sites and subsequent individual mutations to pinpoint key catalytic residues [13], peptide chain region exchange to determine enzymatic specificity [9], and molecular simulation to predict functional regions of elongases and their influence on substrate selectivity [14]. Variations in specific amino acid residues among different elongase types may dictate their unique substrate preferences. Multiple sequence alignments have revealed conserved amino acid differences that correlate with elongase specificity. This study aims to validate the critical role of these amino acid residues through point mutations and molecular modeling.

## 2. Materials and Methods

### 2.1. Sequence Comparison and Structure Prediction

Sequence homology was analyzed using BLAST tools at the National Center for Biotechnology Information (NCBI). Sequence alignment was performed with ClustalW (https://www.ebi.ac.uk/Tools/msa/clustalo/, accessed 8 June 2023). The neighbor-joining method was used to reconstruct the cladogram under the software MEGA11, with the bootstrap value (obtained from 1000 replicates) shown on each node. Analysis of the similarity of fatty acid elongases of different species origin with defined substrate specificity. Initial transmembrane (TM) domain prediction was conducted using the TMHMM program (V2.0, http://www.cbs.dtu.dk/services/TMHMM/, accessed on 15 July 2023).

### 2.2. Strains and Plasmid

*Escherichia coli DH5α* was used for plasmid storage. *Saccharomyces cerevisiae strain INVSc1*, used as a heterologous expression host for functional identification of McELO, PrELO, and PsELO (GenBank accession numbers: OAD03557.1, KAH7468944.1, EGZ16960.1), was purchased from Invitrogen. The pYES2/NT C expression vector, containing the yeast β-galactosidase (GAL1) promoter for inducible protein expression in yeast by galactose, a URA3 auxotrophic marker for selection of yeast transformants, and an ampicillin resistance gene for selection in *E. coli*, was obtained from Invitrogen. The primers used in these steps are listed in Appendix A.

### 2.3. Culture Conditions

*E. coli* was cultivated at 37 °C on LB medium. Yeast was cultured in complete yeast extract peptone dextrose (YPD) medium (1.0 g/L yeast extract, 2.0 g/L Tryptone, 2.0 g/L glucose) at 28 °C. Positive yeast transformants were selected using SC-U selective medium (Synthetic Complete minimum media lacking uracil). To induce gene expression in yeast, an inducing synthetic complete medium without uracil (SC-U medium without glucose, supplemented with 2.0 g/L galactose and 1.0 g/L raffinose) was used. INVSc1 containing pYES2/NT C-ELOs was incubated on SC-U for 48 h at 28 °C, diluted to an OD600 of 0.4, and then an equal amount was added to the induction medium. Cultivation conditions were maintained at 28 °C for 48 h. When needed, 100 μM fatty acid substrates (NU-CHEK, Elysian, MN, USA), including linoleic acid (LA, C18:2^Δ9,12^), α-linolenic acid (ALA, C18:3^Δ9,12,15^), γ-linolenic acid (GLA, C18:3^Δ6,9,12^), arachidonic acid (ARA, 20:4^Δ5,8,11,14^), eicosapentaenoic acid (EPA, 20:5^Δ5,8,11,14,17^), methyl arachidate (mC20:0), and 0.4% Tergitol NP-40 (Sigma, St. Louis, MO, USA, *v*/*v*) for fatty acid solubilization were added to the SC-U induction medium. The cultures were collected and the pellets were used for further analysis.

### 2.4. Yeast Transformation

The McELO, PrELO, and PsELO gene sequences were optimized based on the codon preference of *S. cerevisiae*, synthesized, and cloned into the pYES2/NT C vector (GENEWIZ, Suzhou, China). The resulting plasmids were transformed into *S. cerevisiae* INVSc1 using the PEG/LiAc method. Positive transformants were initially selected on SC-U selective medium.

### 2.5. Immunoblot Analysis of Elongase Expression

The yeast was collected at 6000× *g* for 5 min and resuspended in 10% (*v*/*v*) trichloroacetic acid/acetone containing glass beads and broken using a high throughput tissue breaker and then allowed to settle overnight. Samples were washed with acetone and vacuum dried before resuspension in lysis buffer (8 M urea, 1% (*w*/*v*) sodium dodecyl sulfat (SDS), 10 mM dithiothreitol, 150 mM Tris-HCl, 1% (*v*/*v*) phenylmethanesulfonyl fluoride). The lysate was centrifuged by centrifugation at 10,000× *g*. Proteins were precipitated with acetone and redissolved in 8 M urea before mixing with SDS sample buffer and heating at 95 °C for 5 min for SDS-PAGE and Western blotting. Western blot analysis was performed in the same manner as stated in our prior work [15]. Briefly, 15 μg of cellular protein was separated by SDS-PAGE and transferred to PVDF membranes (Millipore, Burlington, MA, USA). Recombinant proteins were detected using anti-His antibodies.

### 2.6. Mutation Study

Point mutations were introduced by designing primers to amplify the pY-McELO, pY-PrELO, and pY-PsELO vectors, followed by Dpn1 enzyme digestion and transformation into *E. coli*. Mutants Mc Q154A, Pr Y90A, Pr M91A, Pr Y104A, Ps L165A, and Ps Y256A were generated and verified by bidirectional sequencing. The primers used in these steps are listed in Appendix A.

### 2.7. Lipid Extraction and Analysis

Lyophilized yeast cells were disrupted by freeze-thawing in hydrochloric acid and extracted for fatty acids using the chloroform-methanol method, dried by nitrogen blowing, and then methylated by adding 2 mL of 1% (*v*/*v*) sulfuric acid-methanol for 60 min in 80 °C [16]. Fatty acid profiles were analyzed as fatty acid methyl esters (FAMEs) by gas chromatography-mass spectrometry (GC-MS; GCMS-QP2010 Ultra, Shimadzu, Kyoto, Japan). Pentadecanoic acid (C15:0) served as an internal standard, and relative quantification was based on the ratio of each fatty acid to the internal standard peak area. The temperature program was as previously detailed [17]. The transformation efficiency of the mutants was calculated using the substrate conversion rate: Rate of substrate conversion (%) = 100 × [(product)/(product + substrate)], from three independent experiments.

### 2.8. Homology Modelling and Molecular Docking of the Three-Dimensional Structure

Three-dimensional (3D) models of McELO, PrELO, and PsELO were generated using AlphaFold [18]. The refined 3D model geometry was validated using PROCHECK and ERRAT at the SAVES server (SAVESv6.0—Structure Validation Server (https://saves.mbi.ucla.edu, accessed on 15 July 2023)). PROCHECK assessed stereochemical quality, while ERRAT evaluated non-bonded interactions [19]. Molecular docking of protein models with small molecule ligands was performed using AutoDock Vina [20]. The Pymol molecular visualization system (version 2.0, Schrödinger, LLC) was used to visualize protein structures. Mutant amino acid residues were highlighted in different colors. Electrostatic surfaces and cross-sections of McELO, PrELO, and PsELO were generated using PYMOL. Protein stability changes were predicted using the online tool SAAFEC-SEQ [21].

### 2.9. Statistical Analysis

All experiments were conducted with at least three biological replications, with data expressed as mean ± SD. Statistical analysis was performed using IBM SPSS Statistics. A *p*-value < 0.05 was considered statistically significant based on a *t*-test.

## 3. Results

### 3.1. Bioinformatics Analysis of Fatty Acid Elongases McELO, PrELO, and PsELO

McELO, PrELO, and PsELO consist of 276, 267, and 284 amino acids, respectively, and they all are membrane-spanning proteins localized to the endoplasmic reticulum. McELO has six predicted transmembrane domains, while PrELO and PsELO have seven (Appendix A).

We constructed a phylogenetic tree using biochemically characterized elongases from human, algal, yeast, mycobacterial, and oomycete species to examine elongase homology (Figure 1). Based on catalytic substrate classes and phylogenetic trees, elongases fall into four groups [22]: Group I comprises those acting on saturated and/or monounsaturated fatty acids (SFAs/MUFAs), for example, *Nannochloropsis gaditana* (NgΔ0-ELO) that elongates C16:0 to C18:0 [23] and ScELO1-3 from *S. cerevisiae* that catalyzes the elongation of SFAs [24,25]. Group II elongases are Δ9-ELOs, which act on C16:1^Δ9^, C18:2^Δ9,12,^ and C18:3^Δ9,12,15^, including those from *Mortierella alpina* (MaΔ9-ELO) [26] and *Pavlova pinguis* (PpΔ9-ELO) [27]. Groups III and IV comprise Δ6-ELOs and Δ5-ELOs that predominantly elongate C18-Δ6 and C20-Δ5 PUFA, respectively. Phylogenetic analysis showed that McELO belongs to Δ9-ELO, PrELO belongs to Δ6-ELO, and PsELO belongs to SFA/MUFAs-ELO.

Phylogenetic analysis shows that McELO clusters with *M. alpina* Δ9-ELO in a distinct branch separate from algal Δ9-ELOs. PrELO shares greater similarity to Δ6-ELOs from saprophytic moulds compared to other taxa. PsELO exhibits higher homology to algal SFA-ELOs than to human or fungal SFA-ELOs. The SFA-ELO from oomycetes appears uncharacterized thus far (Figure 1).

### 3.2. Functional Analysis of McELO, PrELO, and PsELO Genes in S. cerevisiae

To validate the enzymatic activities of McELO, PrELO, and PsELO, their coding sequences were subcloned into the yeast expression vector pYES2-NT/C and transformed into *S. cerevisiae*, using pYES2 as an empty vector control. PCR amplification confirmed the presence of McELO, PrELO, and PsELO in yeast transformants. The expected amplicon lengths for McELO, PrELO, and PsELO (1140 bp, 1164 bp, and 1197 bp, respectively) matched the results by nucleic acid electrophoresis (Appendix A). Positive transformants were thus named pY-McELO, pY-PrELO, and pY-PsELO. Online ExPASy-ProtParam tool prediction estimated the 6xHis-tagged proteins at 37 kDa. Western blotting revealed 37 kDa antibody-reactive bands in the whole cell extracts of pY-McELO, pY-PrELO, and pY-PsELO transformants, consistent with the predicted size. No bands appeared for the negative control, indicating successful heterologous expression of McELO, PrELO, and PsELO in *S. cerevisiae* (Appendix A).

*S. cerevisiae* naturally contains predominantly long-chain fatty acids C16:0, C16:1^Δ9^, C18:0, C18:1^Δ9^, and C18:1^Δ11^. PY-McELO had significantly lower proportions of C16:0 and C16:1^Δ9^ compared to control, as well as a significant increase in C18:1^Δ9^ content (59.5%) and a little rise in C18:1^Δ11^ (9.3%) (Table 1). The locations of the double bonds on these C18:1 isomers allow us to distinguish their origins: C18:1^Δ11^ was formed from C16:1^Δ9^, while C18:1^Δ9^ was formed from C18:0 by an Δ 9 desaturation [28,29]. Thus, our findings show that the rise in C18:1^Δ9^ and C18:1^Δ11^ caused by McELO expression was due to the activation of the C16:0 and C16:1^Δ9^ elongation reactions. Quantification results suggested that the conversion rate of C16 fatty acids in pY-McELO was converted from 44.9% to 82.4% in the blank control group. PY-PrELO did not show additional peaks compared to the control. Fatty acid analysis of pY-PsELO revealed decreased C16:0 from 20.5% to 15.5%, and new peaks for C20:0 with 2.4% and C22:0 with 0.7% compared to control, suggesting PsELO can sequentially elongate C16:0 to longer chain fatty acids.

PY-McELO fatty acid data showed that McELO could convert 8.4% of LA to C20:2^Δ11,14^ and 3.2% of ALA to C20:3^Δ11,14,17^. Quantification results suggested that PrELO elongated 78. 8% of GLA to DGLA and 1.9% of LA to C20:2^Δ11,14^ (Table 2 and Appendix A). PY-PsELO elongated C20:0 to C22:0 with 37.3% conversion, demonstrating broad substrate specificity. Other FFAs were also tested for McELO, PrELO, and PsELO, but no new products were observed (Appendix A).

### 3.3. McELO, PrELO, PsELO Differentially Conserved Amino Acid Mutation Studies

Fatty acid elongases of the four groups share several domains, including K(X)_5_DT, L(X)_3_HH, N(X)_3_H(X)_2_MYXYY, T(X)_2_Q-(X)_2_Q, and LF(X)_2_F (X, variable amino acid) (Figure 2), indicating that these domains are involved in the elongation activity of the true ELOs. Crystal structures of human ELOVL7 reveal conserved residues coordinate acetyl-CoA binding [30].

Unique conserved amino acids may confer specific substrate preferences on different types of elongases [22]. For instance, YM in box 1 and Y in box 2 occur in ∆6 elongases but not others. In box 3, ∆9-elongases contain LQ while others have LH. In boxes 4 and 5, SFAs/MUFAs elongases harbour unique L and Y residues (Figure 2). However, these structure–function relationships remain hypothetical. To investigate this hypothesis, alanine mutations were performed on conserved residues in McELO, PrELO, and PsELO. To examine whether these conserved sites are critical catalytic sites for elongases, we used the three elongases characterized in this article as templates, alanine mutated the differentially conserved amino acids, heterologously expressed the mutant elongases in yeast, and compared their substrate conversions with those of wild-type and native cultures. Residues mutated were Mc Q154A, Pr Y90A, Pr M91A, Pr Y104A, Ps L165A, and Ps Y256A.

Elongation activity was assessed by adding the corresponding substrate (Figure 3a), and conversion rates varied significantly among mutants. Pr Y90A and Pr M91A converted GLA at only 11%, while Pr Y104A was inactive, well below the 78.8% conversion of the recombinant protein, indicating these three sites significantly impact PrELO catalysis. Ps L165A elongated C20:0 at 15.8% and Ps Y256A lost all C20:0 activity. Compared to the conversion rate of 37.3% for PsELO, the mutant had a significantly lower conversion rate. In contrast, Mc Q154A retained comparable activity to pY-McELO, suggesting this residue is nonessential.

For inactive mutants, Western blots detected protein at levels similar to positives (Figure 3b), implying impaired activity did not result from proteolytic instability. The affected residues were converted from hydrophilic Tyr to hydrophobic Ala, suggesting these amino acids are critical for elongase function.

### 3.4. Three-D Structure Model Demonstrating the Position of the Mutation Site Relative to the Substrate

To interpret the fatty acid conversion capacities of the mutants, protein modeling and molecular docking of McELO, PrELO, and PsELO were carried out. Elongase crystal structures are challenging to obtain due to their transmembrane nature. AlphaFold is a computational approach capable of predicting protein structures to near experimental accuracy in a majority of cases and has been used in membrane proteins [18]. Model quality appeared excellent, with >90% of residues in favoured regions, enabling subsequent molecular docking (Appendix A). McELO, PrELO, and PsELO are all composed of seven transmembrane (TM) helices (TM1-TM7), where TM2-7 form a six TM inverted barrel around a narrow tunnel in which the substrate acyl-coenzyme A docking site is located (Figure 4a).

These mutations were irregularly distributed in the amino acid sequences of the 3D model and the approximate locations were marked in red (Figure 4a). The mutation sites of Pr 90Y, Pr 91M, and Pr 104Y are far away from the substrate action pocket of PrELO, but they can significantly affect the activity of PrELO after mutation. Both Ps 165L and Ps 256Y are near the enzyme’s catalytic pocket, while the phenolic hydroxyl group of Ps 256Y is in close contact with the substrate, and mutating results in the complete inactivity of PsELO. Mc 154Q is within 5 Å of the substrate, but the mutation has no effect on the enzyme activity, and although this site is highly conserved in Δ9-elongase, it is not a critical catalytic site.

The mutants were simulated and showed a decrease in the folding free energy (ΔΔG_fold_), indicating a decrease in thermodynamics mutants stability, which leads to a decrease or loss of activity (Figure 4b). The mutation sites Pr Y90A, Pr M91A, Pr Y104A, Ps165L, and Ps 256Y all have ΔΔG < −0.5 kcal/mol, which affects the thermodynamic stability of the protein. The mutation site Mc Q154A, with ΔΔG > −0.5 kcal/mol in all cases, was recognized as a neutral mutation with negligible effect on protein stability [31]. The fact that overall stability is predicted to be moderately affected supports our results showing conversion capacity of Pr Y90A, Pr M91A, Pr Y104A, Ps165L, and Ps 256Y are decreased.

## 4. Discussion

The diversity of fatty acid elongase catalytic functions across various organisms is closely linked to the differences in fatty acid biosynthetic pathways. Genes encoding polyunsaturated fatty acid elongases have been isolated from a wide range of organisms, including mammals, fungi, lower plants, and algae. ELO-type elongases are phylogenetically classified based on substrate unsaturation positions into SFA-ELO, Δ5-ELO, Δ6-ELO, and Δ9-ELO.

In order to increase the proportion of polyunsaturated fatty acids synthesized by oil-producing microorganisms, higher-activity fatty acid elongases need to be characterized. We characterized three distinct specificities fatty acid elongase from microorganisms containing specific proportions of fatty acids: McELO from *Mucor circinelloides*, PrELO from *Phytophthora ramorum*, and PsELO from *Phytophthora sojae*. Using a heterologous expression system, we analyzed the substrate specificity of these elongases and successfully detected ELO-type elongase expression in *S. cerevisiae* with His tags. Elongases are sparingly expressed endoplasmic reticulum membrane proteins in the *S. cerevisiae* system, and previous studies have not detected their expression using Western Blot.

The identified elongases were classified according to their functions. PY-McELO could significantly reduce the proportion of C16 fatty acids. According to research, McELO has the outstanding catalytic ability for C16 out of the other discovered elongases from various species, including mammals, fungi, and algae [26,29,32]. The percentage of C16 fatty acids in *M. circinelloides* is less than 25%, which may be related to the relatively high catalytic capacity of McELO [33]. Consistent with our findings, it has been reported that the Δ9-elongase (from *M. alpina* and *Yarrowia lipolytica*) also uses C16:0 and C16:1Δ9 as substrates for elongation [6,26]. This is in contrast to other types of Δ9-elongases (from *Physcomitrium patens* and *Chrysochromulina tobinii*) that have higher activity for C18:1Δ9, C18:2Δ9,12. During the production of LC-PUFAs, many strains were engineered to express C16-ELO to increase the target fatty acid content [4,5,6]. Therefore, McELO has great potential for application in genetic engineering modification.

Our discovery indicates that PrELO can catalyze the elongation of GLA to DGLA, thereby demonstrating its Δ6 elongase activity. Among the elongases found in oomycete fungi, filamentous fungi and algae (with GLA conversion ranging from 16–70%), the superiority of PrELO’s GLA conversion capacity (74%) was evident [22,26,34,35,36,37,38,39] (Appendix A). Studies in *M. alpina* indicate the elongation of GLA to DGLA is the rate-limiting step in AA biosynthesis [40,41]. An enzyme that can convert GLA would be useful in attempts to engineer DGLA pathways.

Fatty acid methyl esters analysis of *P. sojae* zoospores showed two long-chain saturated fatty acids (20:0 and 22:0) with 2.1% and 1.5% [42]. PsELO functions as an elongase in *P. sojae*, facilitating the production of long-chain saturated fatty acids. This aligns with its ability to elongate C18:0 to C20:0 and C20:0 to C22:0, as shown in *S. cerevisiae*. The functional characterization results of McELO, PrELO, and PsELO are consistent with the sequence comparison results, thus, by the amino acid sequence we can predict the substrate specificity of the elongase. However, the exact enzyme activity needs to be calculated experimentally.

We introduced point mutations in McELO, PrELO, and PsELO and performed molecular docking to examine the effects on function. We probed the possible roles of conserved residues and found that mutation of Pr Y90, Pr M91, Pr Y104, Ps L165, and Ps Y256 to alanine all inhibited the elongation activity catalyzed by the ELOs but had little effect on protein levels. However, the mutation of Mc Q154 did not affect the elongase activity, and the imidazole group at position 154 was not essential for McELO activity, which is consistent with the conclusion that there was no effect on activity after the H/A mutation at the same position in *Dictyostelium discoideum* EloA [13].

Structural prediction by molecular docking shows Pr Y90 and Pr M91 reside on TM2 distant from the substrate pocket, while Y104 localizes to the loop between TM2 and TM7. Previous reports suggested that the loop between ELOVL-type elongases TM5-7 determines the carbon chain length of the substrate, but the catalytic mechanism targeting the unsaturated bond is unclear [9,30,43]. Molecular docking in this paper shows that the position of the mutated differential amino acid is not in the loop, presumably the tyrosine (Y) residues are negatively charged and cannot bind to Ca^2+^ and Mg^2+^ and thus cannot interact with the phospholipid polar head. The catalytic principle of elongation of enzyme interaction with fatty acid double bonds could not be elucidated, and a more detailed crystal structure is needed to verify it. Moreover, we have predicted that mutations in these particular sites will reduce protein thermodynamic stability, leading to a reduction and loss of catalytic activity.

Computational enzyme engineering advancements have facilitated the rational design of variants with improved catalytic properties [44]. Applying these principles to fatty acid elongase could enhance the yield of mutants. Metabolic engineering using an optimized elongase could increase LC-PUFAs levels in oil-producing microorganisms, providing sustainable ingredients for functional foods.

Overall, this study characterized the specificity of elongation enzymes McELO, PrELO, and PsELO through heterologous expression in *S. cerevisiae*. McELO exhibits strong C16/18 elongase activity, while PrELO demonstrates high Δ6 elongase activity. Mutations to conserved amino acids show that they significantly reduce enzyme activity, and model predictions suggest roles for these amino acids in extending the correct folding of the enzyme. This article provides options for genetically engineered modifications and enhances our understanding of elongase catalytic properties.

## Figures and Tables

**Figure 1 jof-10-00129-f001:**
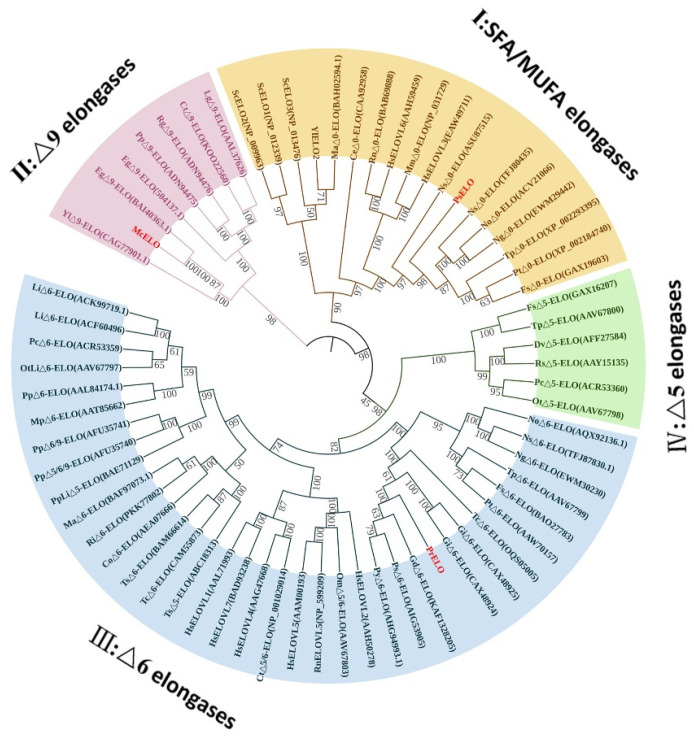
The GenBank ID of the elongases is indicated in the brackets. Elongase McELO, PrELO, PsELO are marked in red. Co, *Conidiobolus obscurus*; Ct, *Chrysochromulina tobinii*; Dv, *Diacronema viridis*; Fs, *Fistulifera solaris*; Gs, *Globisporangium splendens*; Hs, *Homo sapiens*; Ig, *Isochrysis galbana*; Li, *Lobosphaera incisa*; Ma, *Mortierella alpina*; Mc, *Mucor circinelloides*; Mi, *Myrmecia incisa*; Mm, *Mus musculus*; Mp, *Marchantia polymorpha*; Ng, *Nannochloropsis gaditana*; Ns, *Nannochloropsis salina*; Ot, *Ostreococcus tauri*; Pc, *Pyramimonas cordata*; Pp, *Physcomitrium patens*; Ppi, *Pavlova pinguis*; Pr, *Phytophthora ramorum*; Ps, *Phytophthora sojae*; Psp, *Pavlova sp*; Pt, *Phaeodactylum tricornutum*; Ri, *Rhizophagus irregularis*; Rn, *Rattus norvegicus*; Rs, *Rebecca salina*; Tc, *Thraustotheca clavata*; Tp, *Thalassiosira pseudonana*; and Ts, *Thraustochytrium sp*; Yl, *Yarrowia lipolytica*.

**Figure 2 jof-10-00129-f002:**
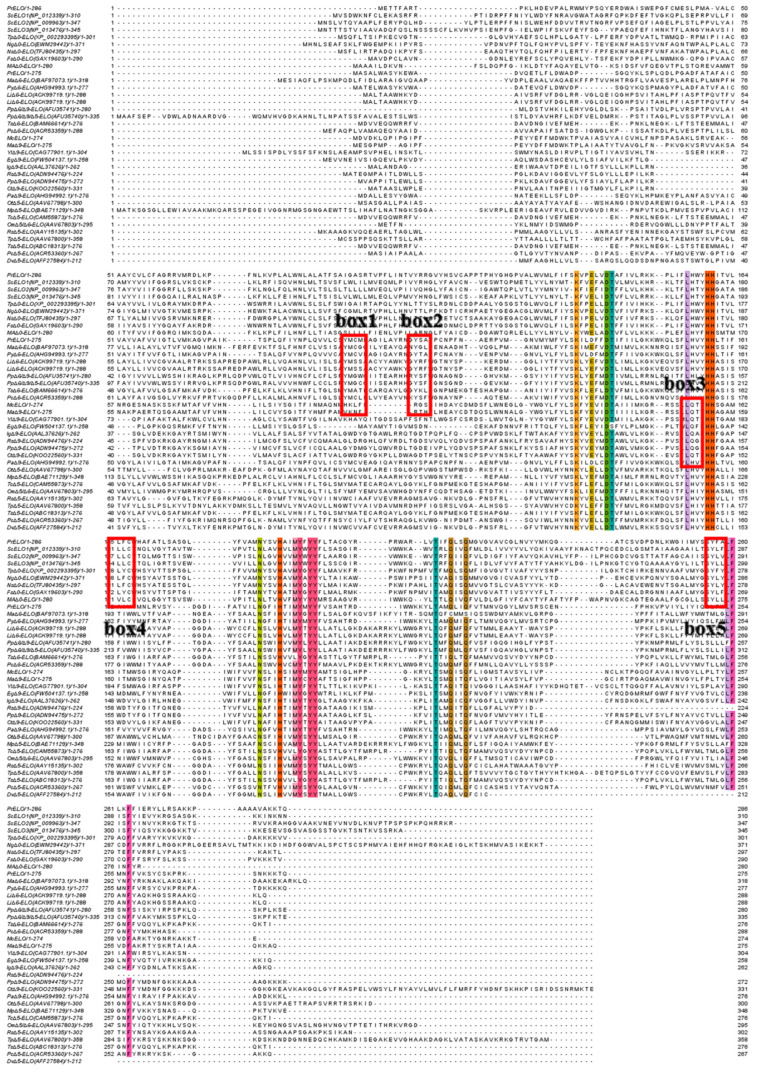
Protein sequence comparison of fatty acid elongases. Multiple sequence comparison was performed using the online software ClustalW (https://www.ebi.ac.uk/Tools/msa/clustalo/, accessed 8 June 2023) with default parameters. For more information on protein names and their IDs, see Figure 1 legend.

**Figure 3 jof-10-00129-f003:**
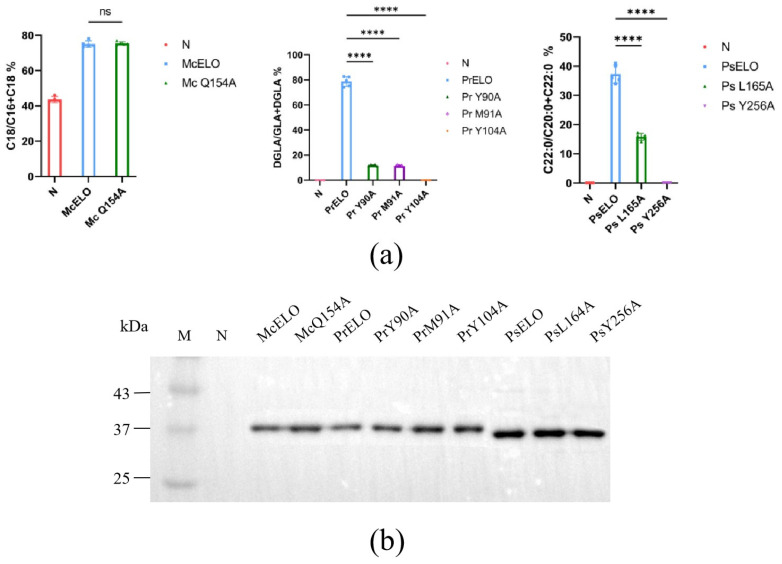
Validation of protein expression and substrate conversion of mutant transformants. (**a**) Rate of substrate conversion of the control, the recombinant protein, and the mutants in *S. cerevisiae* incubated with 100 μM LA, GLA, or C20:0 as substrates for 24 h at 28 °C. Rate of substrate conversion = 100 × [(product)/(product + substrate)]. (**b**) Immunoblot analysis of yeast expressing mutant McELO, PrELO, and PsELO alleles. The significant difference is indicated by **** *p* < 0.0001. No significant difference is indicated by ns.

**Figure 4 jof-10-00129-f004:**
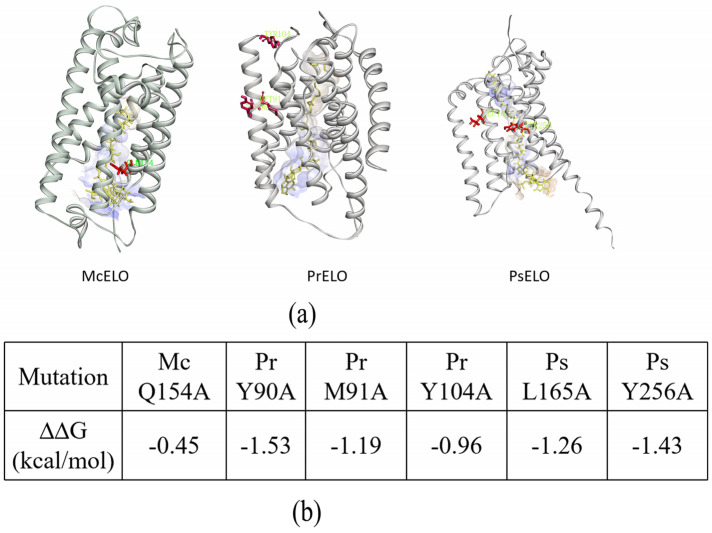
(**a**) The 3D structures and central acyl-CoA-binding tunnels of the elongase proteins of McELO, PrELO, and PsELO were predicted by homology modeling and molecular docking. The relative positions of each mutation site with red marker in the elongase structure. (**b**) The change in protein stability was calculated using SAAFEC-SEQ, with the negative free energy variation indicating a loss of protein stability.

**Table 1 jof-10-00129-t001:** Fatty acid composition (% *w*/*w*) of *S. cerevisiae* transformants overexpressing McELO, PrELO, or PsELO.

	Fatty Acid Composition (%)								Dry Cell Weight (g/L)	Lipid Content (%)
	C16:0	C16:1^Δ9^	C18:0	C18:1^Δ9^	C18:1^Δ11^	C20:0	C20:1^Δ11^	C22:0		
control	20.6 ± 0.5 ^a^	34.4 ± 0.7 ^a^	12.8 ± 0.3 ^c^	29.8 ± 0.4 ^b^	2.4 ± 0.07 ^b^	ND	ND	ND	2.04 ± 0.1	9.70 ± 0.3
+McELO	3.8 ± 0.2 ^c^	7.4 ± 0.1 ^b^	15.6 ± 0.2 ^b^	60.1 ± 0.6 ^a^	7.5 ± 0.2 ^a^	ND	5.6 ± 0.2	ND	2.56 ± 0.2	9.50 ± 0.4
+PrELO	21.6 ± 1.3 ^a^	36.2 ± 1.9 ^a^	12.0 ± 0.7 ^c^	27.5 ± 1.5 ^b^	2.6 ± 0.3 ^b^	ND	ND	ND	2.18 ± 0.2	8.34 ± 0.3
+PsELO	15.5 ± 0.8 ^b^	34.5 ± 2.3 ^a^	17.7 ± 0.4 ^a^	27.1 ± 0.9 ^b^	2.3 ± 0.1 ^b^	2.4 ± 0.3	ND	0.7 ± 0.1	2.05 ± 0.1	8.25 ± 0.8

Values are mean of three samples ± standard error of the mean. Values with the same letter in the same column are not significantly different. a, *p*-value < 0.05, b, *p*-value < 0.01, c, *p*-value < 0.001. ND: not detected.

**Table 2 jof-10-00129-t002:** Fatty acid composition (% *w*/*w*) of control, pY-McELO, pY-PrELO, or pY-PsELO with fatty acid substrates of LA, GLA and C20:0.

	Fatty Acid Composition (%)									
	C16:0	C16:1	C18:0	C18:1	C18:2	C18:3	C20:0	C20:3	C20:2	C22:0
Control + C18:2^Δ9,12^	24.7 ± 0.8	30.6 ± 3.1	8.3 ± 1.9	26.7 ± 2.3	9.6 ± 1.3	-	-	-	-	-
McELO + C18:2^Δ9,12^	6.9 ± 0.2	12.3 ± 2.1	13.5 ± 0.3	57.4 ± 3.1	8.9 ± 1.6	-	-	-	0.9 ± 0.2	-
Control + C18:3^Δ9,12,15^	23.6 ± 0.7	30.8 ± 1.5	7.8 ± 0.6	25.9 ± 1.7	-	11.9 ± 0.4	-	-	-	-
PrELO + C18:3^Δ9,12,15^	24.8 ± 1.3	30.5 ± 1.1	8.7 ± 0.4	23.8 ± 1.7	1.4 ± 0.2	2.8 ± 0.5	-	8.0 ± 0.6	-	-
Control + C20:0	24.2 ± 1.1	30.0 ± 2.2	10.9 ± 1.2	28.5 ± 2.5	-	-	6.4 ± 0.6	-	-	-
PsELO + C20:0	18.9 ± 0.5	35.5 ± 1.2	6.8 ± 0.3	27.0 ± 0.8	-	-	8.7 ± 0.2	-	-	3.1 ± 0.2

## Data Availability

The datasets that support the findings of the current study are available from the corresponding author upon reasonable request.

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
