# Peer review of "Characterization of Three Types of Elongases from Different Fungi and Site-Directed Mutagenesis"

_jof, 2024, doi:10.3390/jof10020129_

Round 1
Reviewer 1 Report
Comments and Suggestions for Authors
Manuscript by Wang et al. entitled “Characterization of three types of elongases from different fungi and site-directed mutagenesis“ studies function of three fatty acid elongases from fungi. The topic of the study is certainly interesting and relevant for readers. The results are interesting, however, presentation of the results is not completely clear in some cases. Therefore, I recommend revision of the manuscript, especially:
- The description of the calculation of substrate conversion in section 2.6 should be moved rather to section 2.7 or on another suitable place.
- Methylation procedure should be explained in detail or referenced (line 135).
- In line 220, there is a calculation of conversion rate equation again. However, in the Table there are no such data.
- Description of the Table 2 is not sufficient. It should be noted that external fatty acids were supplemented to media.
- I do not understand the meaning of the integer numbers in Table 2. Also, I miss the data for GLA supplementation in the Table 2.
- Sums of fatty acids in Table 1 and Table 2 are not 100 %. What is the missing fatty acid (although it is less than 1%)?
- In Figure 3 description there is reversed (a) for (b) by mistake.
Comments on the Quality of English Language- The language correction would be beneficial, there are several mistakes/typing errors in the manuscript (e. g. lines 35, 42, 47, 271).
Author Response
Reviewer 1:
Manuscript by Wang et al. entitled "Characterization of three types of elongases from different fungi and site-directed mutagenesis" studies function of three fatty acid elongases from fungi. The topic of the study is certainly interesting and relevant for readers. The results are interesting, however, presentation of the results is not completely clear in some cases.
Authors’ Response: Thank you for your time and efforts in reviewing our manuscript, we all greatly appreciate it.
Comment 1: The description of the calculation of substrate conversion in section 2.6 should be moved rather to section 2.7 or on another suitable place.
Response: Thank you for your valuable comments. The calculation of fatty acid substrate conversion is more appropriately placed in the Lipid extraction and analysis, and based on your suggestion, we have moved section 2.6 to section 2.7.
Comment 2: Methylation procedure should be explained in detail or referenced (line 135).
Response: Thank you for your advice. We have added fatty acid methyl esterification references to the manuscript.
Revision: Line 133-135
Lyophilized yeast cells were disrupted by freeze-thawing in hydrochloric acid and extracted for fatty acids using the chloroform-methanol method, dried by nitrogen blowing, and then methylated by adding 2 ml of 1% (v/v) sulfuric acid-methanol for 60 min in 80°C.
Metcalfe, L.D.; Schmitz, A.A.; Pełka, J.J.A.C. Rapid Preparation of Fatty Acid Esters from Lipids for Gas Chromatographic Analysis. 1966, 38, 514-515.
Comment 3: In line 220, there is a calculation of conversion rate equation again. However, in the Table there are no such data.
Response: Thank you for your comments. We have removed the calculation of the conversion rate equation from Table 2.
Comment 4: Description of the Table 2 is not sufficient. It should be noted that external fatty acids were supplemented to media.
Response: Thank you for your advice about the description of Table 2. We have corrected the title of Table 2 based on your suggestion to "Fatty acid composition (% w/w) of control, pY-McELO, pY-PrELO, or pY-PsELO with fatty acid substrates of LA, GLA and C20:0".
Comment 5: I do not understand the meaning of the integer numbers in Table 2. Also, I miss the data for GLA supplementation in the Table 2.
Response: Thank you for your advice. The values in Table 2 indicate the percentage of total lipids accounted for by each type of fatty acid, and the percentages of some fatty acids are integer numbers. GLA in table 2 is expressed as C18:3Δ9,12,15.
Comment 6: Sums of fatty acids in Table 1 and Table 2 are not 100 %. What is the missing fatty acid (although it is less than 1%)?
Response: Thanks for pointing these problems out. This was a mistake in our calculations, and we have revised the data in the table to ensure that the fatty acids in each group summed to 100%.
Comment 7: In Figure 3 description there is reversed (a) for (b) by mistake.
Response: Thanks for pointing these problems out. We have updated the graphical notation for Figure 3.
Comment 8: The language correction would be beneficial, there are several mistakes/typing errors in the manuscript (e. g. lines 35, 42, 47, 271).
Response: Thank you for your comments. We were really sorry for our careless mistakes. We have revised and corrected the English grammar and sentence structure throughout the manuscript. The changes were marked in red in the revised manuscript.
Revision:
-Line 35-36
LC-PUFAs biosynthesis still encounters many challenges, such as the lack of converting enzymes with higher activity and how to break through the limitation of lipid homeostasis.
-Line 41-44
The elongase system is composed of four enzymes: a condensing enzyme (β-ketoacyl CoA synthase, KCS), β-ketoacyl CoA reductase (KCR), β-hydroxyacyl CoA dehydrase, and trans-2-enoyl CoA reductase.
-Line 47-48
Highly activity elongases can be isolated from microorganisms rich in polyunsaturated fatty acids or with specific fatty acid ratios.
Line 272-273
To interpret the fatty acid conversion capacities of the mutants, protein modeling and molecular docking of McELO, PrELO, and PsELO were carried out.
Reviewer 2 Report
Comments and Suggestions for Authors
Review
for the article titled “Characterization of three types of elongases from different fungi and site-directed mutagenesis” by Wang et al.
The article presents new data on three types of elongases from different fungal species and site-directed mutagenesis. It is suitable for publication in the JoF journal. However, the minor revision is required.
Point 1: Keywords, line 24, 25 – include the elongase-producing fungal species
Point 2: page 1, line 30 – List natural sources of long-chain polyunsaturated fatty acids in parentheses.
Author Response
The article presents new data on three types of elongases from different fungal species and site-directed mutagenesis. It is suitable for publication in the JoF journal. However, the minor revision is required.
Authors’ Response: Thank you for your time and efforts in reviewing our manuscript, we all greatly appreciate it.
Comment 1: Keywords, line 24, 25 – include the elongase-producing fungal species
Response: Thank you for your comment. We have added the keywords with the elongase-producing fungal species.
Revision: Line 24-25
Fatty acid elongase, Characterization, Mucor circinelloides, Phytophthora ramorum, Phytophthora sojae, Mutagenesis
Comment 2: page 1, line 30 – List natural sources of long-chain polyunsaturated fatty acids in parentheses.
Response: Thank you for your advice. We list natural sources of long-chain polyunsaturated fatty acids in parentheses.
Revision: Line 30-31
The limited natural sources of LC-PUFAs (fish oil and algae oil) cannot satisfy the growing market demand.